# Efficient support of virus-like particle assembly by the HIV-1 packaging signal

**Mauricio Comas-Garcia[†‡], Tomas Kroupa, Siddhartha AK Datta, Demetria P Harvin, Wei-Shau Hu, Alan Rein\***

HIV Dynamics and Replication Program, Center for Cancer Research, National Cancer Institute, Frederick, United States

**Abstract** The principal structural component of a retrovirus particle is the Gag protein. Retroviral genomic RNAs contain a 'packaging signal' ('Ψ') and are packaged in virus particles with very high selectivity. However, if no genomic RNA is present, Gag assembles into particles containing cellular mRNA molecules. The mechanism by which genomic RNA is normally selected during virus assembly is not understood. We previously reported (*Comas-Garcia et al., 2017*) that at physiological ionic strength, recombinant HIV-1 Gag binds with similar affinities to RNAs with or without Ψ, and proposed that genomic RNA is selectively packaged because binding to Ψ initiates particle assembly more efficiently than other RNAs. We now present data directly supporting this hypothesis. We also show that one or more short stretches of unpaired G residues are important elements of Ψ; Ψ may not be localized to a single structural element, but is probably distributed over >100 bases.

DOI: https://doi.org/10.7554/eLife.38438.001

**\*For correspondence:**
reina@mail.nih.gov

**Present address:** [†]Facultad de Ciencias, Universidad Autónoma de San Luis Potosí, San Luis Potosí, México; [‡]Centro de Investigaciones en Ciencias de la Salud y Biomedicina, Universidad Autónoma de San Luis Potosí, San Luis Potosí, México

**Competing interests:** The authors declare that no competing interests exist.

## Introduction

A retrovirus particle is assembled from ~1500–3000 molecules of the Gag protein, together with RNA (*Vogt and Simon, 1999*), as well as smaller amounts of other viral and cellular proteins and a surrounding lipid bilayer. In a cell infected with wild-type virus, the vast majority of the released particles contain the genomic RNA (gRNA) of the virus, despite the fact that this RNA is only a minor species in the virus-producing cell (*Chen et al., 2009*). The selection of the gRNA for encapsidation depends upon the presence in this RNA of the 'packaging signal' or 'Ψ', a region of ~200 or more bases near the 5' end of the viral RNA (*Aldovini and Young, 1990*; *Berkowitz et al., 1996*; *Comas-Garcia et al., 2016*; *D'Souza and Summers, 2005*). However, the nature of Ψ and the mechanism of selective packaging of gRNA are not well understood as yet.

In mammalian cells expressing Gag in the absence of Ψ-containing RNA, the protein assembles into virus-like particles (VLPs) structurally indistinguishable from immature virions; these particles contain roughly the same amount of RNA as wild-type particles, but this RNA is a nearly random sample of cellular mRNA molecules (*Rulli et al., 2007*). Similarly, recombinant Gag protein can assemble into VLPs in a defined system in vitro; while this assembly requires the presence of RNA (or DNA), virtually any single-stranded nucleic acid can support assembly under these conditions (*Campbell et al., 2001*; *Campbell and Rein, 1999*).

In an effort to understand the selective packaging of Ψ-containing RNA, we recently measured the affinity of recombinant HIV-1 Gag protein (lacking the p6 domain at its C-terminus) for different RNAs (*Comas-Garcia et al., 2017*). We found that the protein has similar, very high affinities for all the RNAs tested when assayed at near-physiological ionic strengths. However, further examination showed that this affinity is the sum of both specific and non-specific interactions. Non-specific binding could be selectively reduced by mutating specific residues in the protein; or by adding a vast excess of an irrelevant competitor RNA; or simply by raising the ionic strength in the assay. When

the binding measurements were modified in any of these ways, a strong specific interaction with Ψ could be detected. The salt-resistance of the binding of Gag to Ψ had previously been observed, using somewhat different techniques, by Webb et al. (*Webb et al., 2013*).

To explain how Ψ-containing RNAs are selectively packaged, despite the fact that Gag binds any RNA tightly at physiological ionic strength and any RNA can support assembly, we proposed that binding to Ψ leads to initiation of assembly more efficiently than binding to other RNAs (*Comas-Garcia et al., 2016*; *Nikolaitchik et al., 2013*). We now present in vitro data that lend strong support to this hypothesis. This work also includes a preliminary characterization of the RNA sequences that are specifically bound by Gag under the modified assay conditions described above.

Gag has been suggested to bind specifically to several distinct sites in the 5' region of HIV-1 RNA (*Lever, 2007*). These include an internal loop and surrounding bases in stem-loop 1, the locus of the 'kissing interaction' where dimerization of gRNA is initiated (*Abd El-Wahab et al., 2014*); stem-loop 2 (*Amarasinghe et al., 2000*); stem-loop 3 (sometimes called 'Ψ') (*De Guzman et al., 1998*); and a series of very short unpaired stretches, each with one or more unpaired G residues, collectively termed the 'Nucleocapsid Interaction Domain' (*Wilkinson et al., 2008*). We tested several of these possibilities by introducing mutations into a 'Ψ' construct and testing the binding of Gag under different conditions.

## Results and discussion

One important unresolved question is the exact sequence(s) which define Ψ. We measured binding affinities using, where not specified otherwise, the methodologies described earlier (*Comas-Garcia et al., 2017*), except that the RNAs were 401 bases in length rather than 190 nts. These RNAs begin at either nt 150 or nt 200 (see *Figure 1A*) and were labeled at their 3' ends with Cy5. As indicated in the Figure, the mutants included individual deletions spanning either stem-loop 1 or stem-loop 3, and the 'Multiple Binding Site Mutant' (MBSM), in which all of the G's in the stretches identified by Wilkinson et al. as the Nucleocapsid Interaction Domain (*Wilkinson et al., 2008*) were replaced with A's. We also noted that these RNAs contain a run of unpaired G and C residues (nt 442–459) that may well be paired in full-length RNA, but not in our 401-base RNAs. To test the possibility that these bases contribute to specific binding of Gag to the transcripts, we also mutated these residues to A's, both in the otherwise wild-type construct beginning at nt 200 (creating the 'GC loop mutant') and in the MBSM; this construct is designated 'MBSM second generation'. In all cases, removal of bases by deletion was compensated by extending the 3' end of the RNA, so that all the RNAs were 401 bases long. As a negative control RNA, we produced the reverse complement of Ψ, that is RNA complementary to nt 200–600.

To test the effects of these changes upon the specific and non-specific binding by Gag, we titrated Gag into these RNAs, monitoring binding by the quenching of the fluorophore as described (*Comas-Garcia et al., 2017*), either in binding buffer (containing 0.2M NaCl), or in binding buffer with a 50-fold excess by mass of yeast tRNA, or in binding buffer containing 0.4M NaCl. Results of these assays are shown in *Figures 1B, C and D*, respectively. It is evident that Gag binds all the tested RNAs well in binding buffer. However, addition of yeast tRNA (*Figure 1C*) or raising the ionic strength in the assay (*Figure 1D*) strongly depressed binding to both iterations of the MBSM RNA, while deleting either SL1 or SL3 did not. Binding to the reverse complement RNA was drastically reduced under both of these conditions. These results show that the specific binding of Gag to Ψ, detected in these assays, depends upon some or all of the clusters of unpaired G residues called the Nucleocapsid Interaction Domain (*Wilkinson et al., 2008*), but neither stem-loop one nor stem-loop three is crucial for this binding (*Figure 1C and D*). A similar mutant has been reported to be deficient in selective packaging in vivo (*Keane et al., 2015*).

As discussed above, we have proposed that genomic RNA is selectively packaged because binding to Ψ is particularly efficient at initiating VLP assembly (*Comas-Garcia et al., 2016*). Thus, it was of interest to assess the abilities of the different RNAs to support VLP assembly. For these experiments we focused on the Ψ that starts at nt 200, the MBSM second generation and the Reverse complement RNA. Also, for these experiments on particle assembly, we used Gag protein lacking most of the matrix (MA) domain, as well as p6: we have previously reported that Δp6 assembles into VLPs with radii of curvature drastically different from those of authentic virions (*Campbell and Rein, 1999*), indicating that they are quite different in overall structure from authentic immature particles.

In contrast, the deleted protein ('Δ16–99 Gag', also frequently called 'ΔMA') assembles into VLPs in which the lattice of proteins closely mimics that in immature HIV-1 virions (*Campbell et al., 2001*; *Briggs et al., 2009*; *Fäcke et al., 1993*; *Gross et al., 2000*; *Wilk et al., 2001*).

We first compared the binding to RNA of Δ16–99 Gag with that of Gag. Our previous measurements monitored RNA-binding using the ability of Gag to quench the Cy5 fluorophore on the RNA (*Comas-Garcia et al., 2017*). However, we found that Δ16–99 Gag does not quench the fluorophore; evidently, the quenching involves the MA domain. Therefore, we used microscale thermophoresis (MST) for monitoring binding by this protein. As shown in *Figure 2A*, MST and quenching measurements give very similar results for the binding of Δp6 Gag to the Ψ RNA that starts at nt 200; at 0.15 M NaCl the $K_D$s for MST and FCS were 14 and 17 nM, respectively, while at 0.45 M NaCl they were 256 and 302 nM. In all cases the Hill coefficient was greater than 1.0. These data show that MST is able to recapitulate our original FCS results (*Comas-Garcia et al., 2017*). MST data are presented in more detail in *Figure 2—figure supplement 1* and *Table 1*. Interestingly, Δ16–99 Gag bound relatively weakly to all 3 RNAs at 0.5M NaCl (see *Table 1*). The implications of this result are now under further investigation.

We wished to quantitatively compare the different RNAs with respect to their ability to support assembly. It was important that all of the RNA be bound by the Δ16–99 Gag protein in these

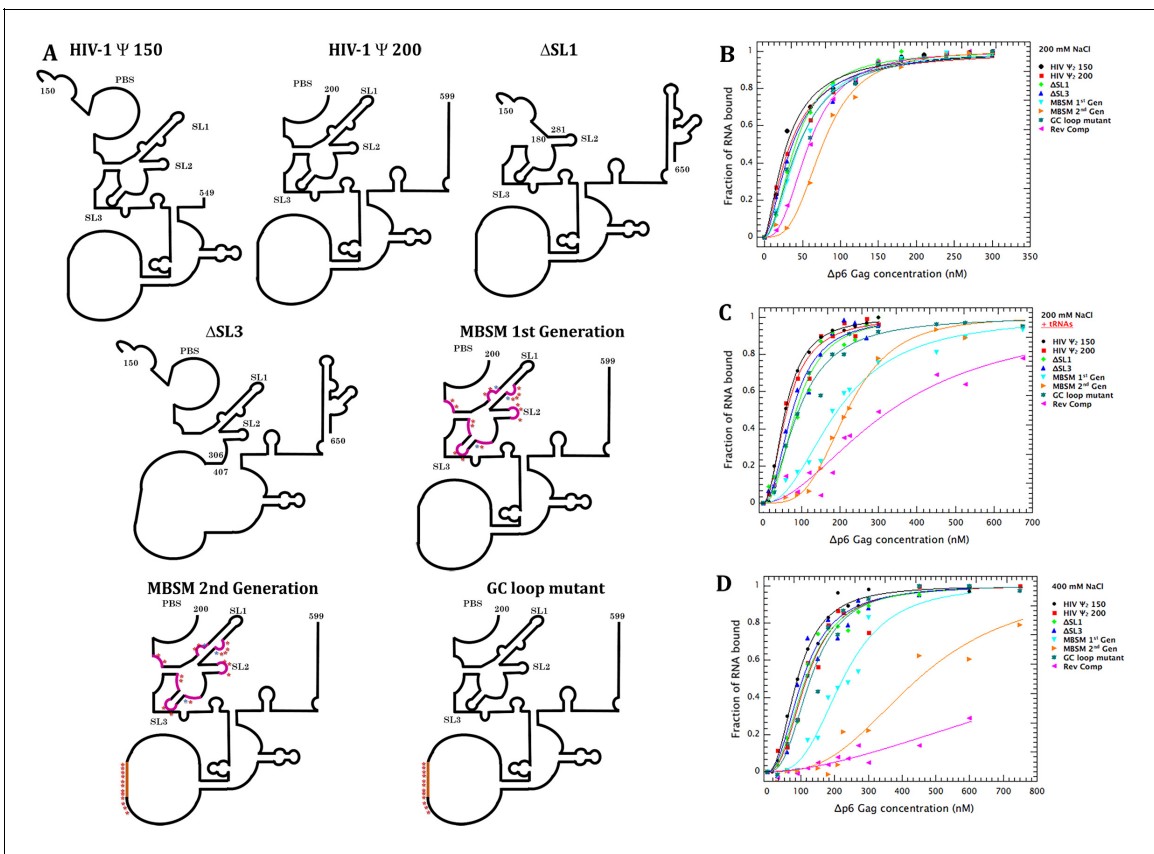

**Figure 1.** Schematic representation of the tested RNAs and their binding profiles to Gag measured by Cy5 quenching. (**A**) Schematic representation of the expected secondary structure of the RNAs used in these experiments. These representations are based on the secondary structure proposed by Wilkinson and co-workers (*Wilkinson et al., 2008*). The purple stars in the MBSM first and second generation and the GC loop mutant RNAs indicate mutations of G to A, while the blue stars represent C to A mutations. (**B**) Binding curves for all of the tested RNAs with Δp6 Gag at 200 mM NaCl, monitored by quenching as previously described (*Comas-Garcia et al., 2017*). The buffer in this assay contained 0.2M NaCl, 20 mM Tris-HCl pH 7.5, 5 mM MgCl₂, 1 µM ZnCl₂, 0.1 mM PMSF, 1 mM β-mercaptoethanol, and 0.05%(v/v) Tween 20. (**C**) Binding curves obtained as in (**B**), but in the presence of a 50-fold excess by mass of yeast tRNA. (**D**) Binding curves obtained as in (**B**), but in a buffer containing 400 mM, rather than 200 mM, NaCl. Values in (**B–D**) are means of two independent experiments, and each point in each experiment is the mean of 10 measurements. Experiments giving $K_D$ values differing by >10% from the consensus values were discarded.

DOI: https://doi.org/10.7554/eLife.38438.002

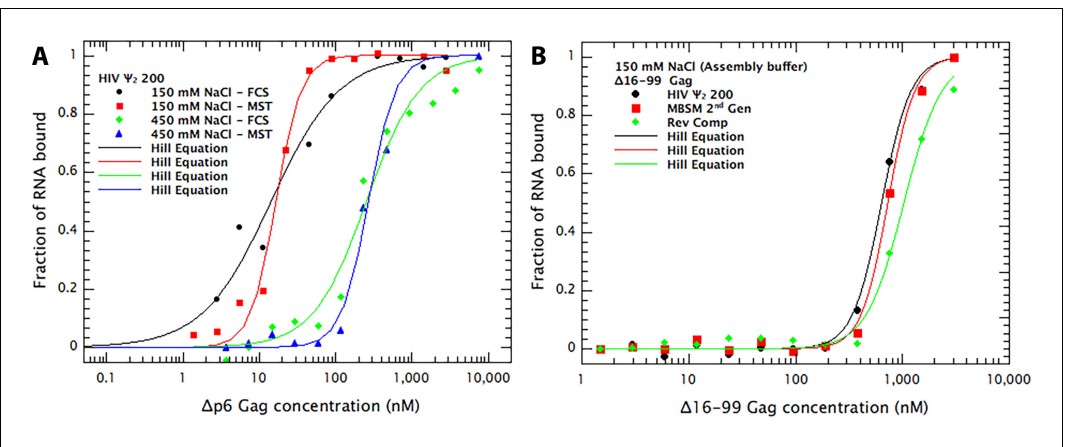

**Figure 2.** Comparison of RNA-Gag binding measurements by Cy5 quenching and Microscale Thermophoresis (MST). (**A**) Comparison of FCS (i.e. Cy5 quenching in FCS apparatus) and MST methods for measurement of binding of Δp6 Gag to dimeric Ψ 200 RNA. The Cy5-tagged RNA was dimerized as described (**Comas-Garcia et al., 2017**) and diluted into binding buffer B to a concentration of 7 nM. This buffer was composed of 50 mM phosphate, pH 7.0, 0.05% Tween 20, 0.1 mM PMSF, and 1 mM β-mercaptoethanol, together with either 0.15 M or 0.45 M NaCl. The sample was then divided and, after 16 hr at 4°C, used for binding measurements by FCS or MST. Both methods give very similar $K_D$s, although the MST curves suggest somewhat higher cooperativity in the binding than FCS. (**B**) Binding of Δ16–99 Gag protein to the three RNAs used for the Virus-like-particle (VLP) assembly experiments. Ψ 150 RNA, MBSM second generation RNA, and Reverse Complement RNA were all treated as described (**Comas-Garcia et al., 2017**) for Ψ dimerization. They were then diluted into Assembly Buffer (20 mM Tris pH 7.5, 0.15M NaCl, 5 mM MgCl$_2$, 1 μM ZnCl$_2$, 0.1% Tween 20, 0.1 mM PMSF, and 1 mM DTT). Binding of Δ16–99 Gag to the RNAs was then measured by MST. The FCS data in **Figure 2A** was treated as in **Figure 1B–D**. All MST data results are the means of three independent experiments. Each data-point in each MST experiment is the mean of triplicate measurements.

DOI: https://doi.org/10.7554/eLife.38438.004

The following figure supplement is available for figure 2:

**Figure supplement 1.** MST data on binding of Δ16–99 Gag protein to ψ RNA.

DOI: https://doi.org/10.7554/eLife.38438.005

experiments, so that any differences observed represent differences in support of assembly, not differences in the extent of binding. **Figure 2B** shows the results of MST binding assays in a buffer closely resembling that used in assembly experiments, yielding $K_D$s of 226, 382, and 568 nM for Ψ (beginning at nt 200), MBSM second Generation (Gen), and Reverse Complement (Rev Comp) RNAs, respectively. Specifically, it is evident that nearly all of each RNA is bound at 1–2 μM Δ16–99 Gag, significantly below the levels used in the assembly experiments (see **Figure 3** below).

**Table 1.** Results of MST measurements of binding of Δ16-99 Gag to RNAs at 0.15 and 0.5M NaCl. The Table shows means and standard deviations of replicate measurements.

| RNA (0.15M NaCl) | $K_d$ (nM) | Error | $n_H$ | Error |
|---|---|---|---|---|
| Ψ200 | 645 | 16 | 3.2 | 0.2 |
| MBSM 2$^{nd}$ gen | 737 | 17 | 3.5 | 0.3 |
| Rev Comp | 1042 | 42 | 2.5 | 0.2 |
| RNA (0.5M NaCl) | | | | |
| Ψ200 | 945 | 267 | 1.4 | 0.1 |
| MBSM 2$^{nd}$ gen | 2200 | 151 | 1.1 | 0.1 |
| Rev Comp | 2479 | 109 | 1.3 | 0.1 |

DOI: https://doi.org/10.7554/eLife.38438.003

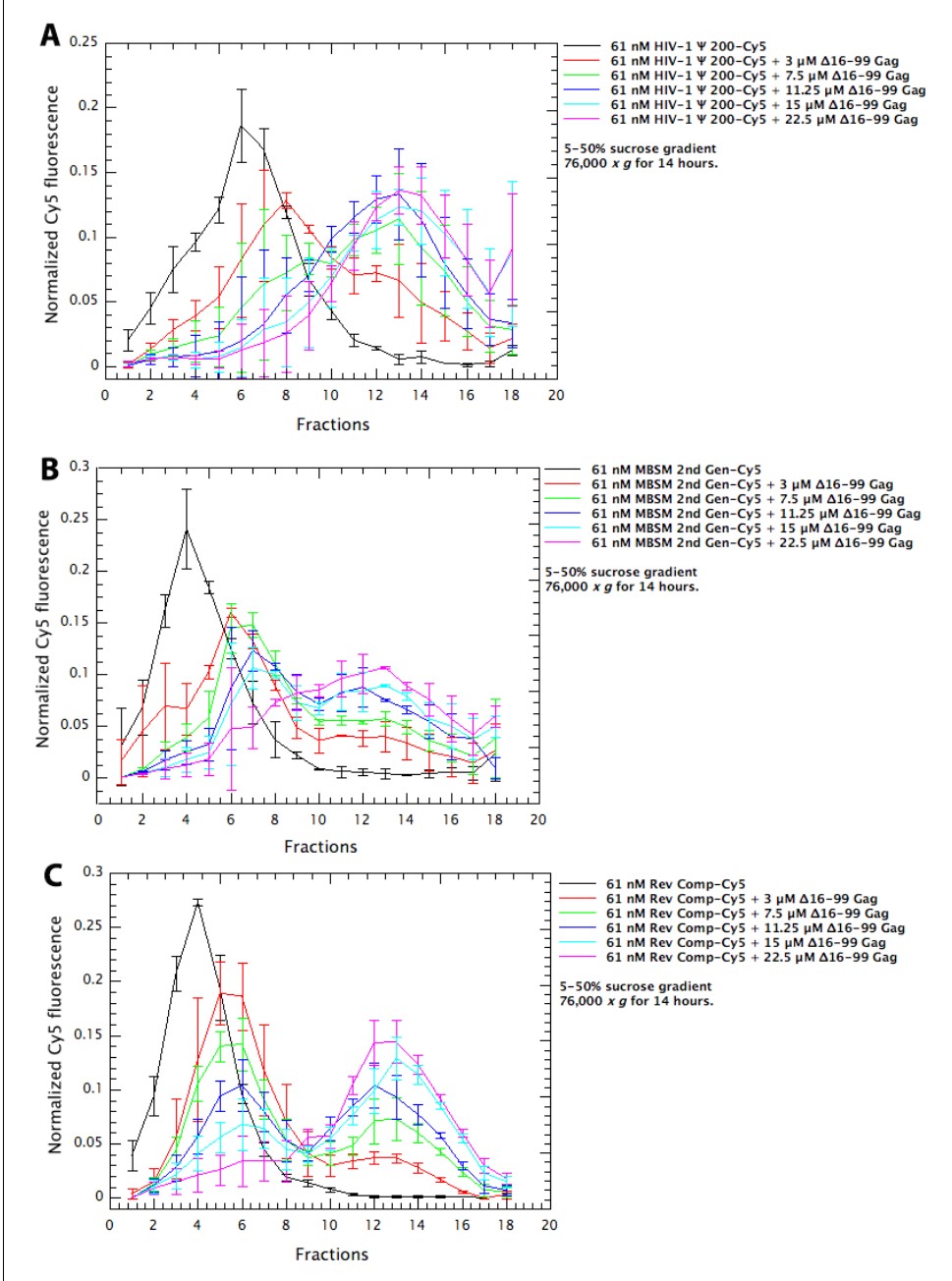

**Figure 3.** Assembly of Δ16–99 Gag protein on different RNAs. Cy5-tagged Ψ 200 RNA (panel A), MBSM second generation RNA (panel B), and Reverse Complement RNA (panel C) were all treated as in the 'RNA Dimerization' protocol (*Comas-Garcia et al., 2017*). They were then diluted to 61 nM in Assembly Buffer and Δ16–99 Gag was titrated into these solutions. After 6 hr at 4°C, the mixtures were layered on 5–50% (w/v) sucrose gradients. The gradients had the same composition as Assembly Buffer except that they did not contain Tween 20, β-mercaptoethanol, or PMSF. After centrifugation for 14 hr at 76,000 x g, fractions were collected from top to bottom and assayed for Cy5 fluorescence and for Gag protein content by spotting aliquots onto nitrocellulose membrane and immunoblotting with anti-p24$^{CA}$ antiserum. The points are means and standard deviations of 3 independent experiments; experiments were excluded if the positions of the peaks were different from these consensus profiles.

DOI: https://doi.org/10.7554/eLife.38438.006

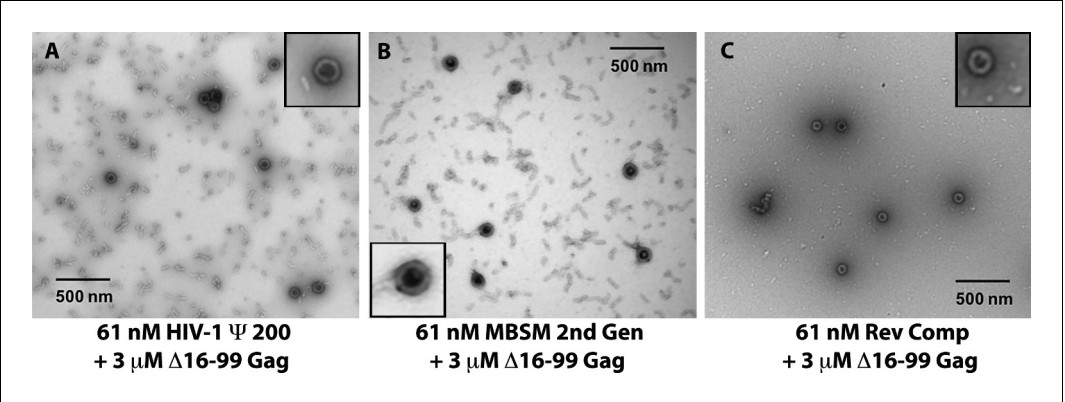

**61 nM HIV-1 Ψ 200**
**+ 3 μM Δ16-99 Gag**

**61 nM MBSM 2nd Gen**
**+ 3 μM Δ16-99 Gag**

**61 nM Rev Comp**
**+ 3 μM Δ16-99 Gag**

**Figure 4.** Negative stain electron micrographs on aliquots from the assembly reactions in (A-C). Insets: well-formed VLPs at higher magnification.

DOI: https://doi.org/10.7554/eLife.38438.007

Finally, we compared different RNAs with respect to their ability to support assembly of Δ16–99 Gag into VLPs. Different amounts of Δ16–99 Gag were added to 61 nM solutions of the Cy5-labeled RNAs; VLP assembly was monitored by the shift of the RNA into large, rapidly sedimenting structures, and was confirmed by negative-stain electron microscopy (*Figure 4*). Although well-formed VLPs were visible in all the reactions (see insets in the Figure), a variety of other structures were also observed, particularly in the ψ and MBSM samples. The mixtures were layered onto sucrose gradients and centrifuged at 76,000 x g for 14 hr. Fractions were collected and assayed for both Cy5 fluorescence and Δ16–99 Gag protein content ($p24^{CA}$ signal). Results of this experiment for Ψ, MBSM second Gen, and Rev Comp RNAs are shown in *Figure 3A–C*. In each panel, the black line is the sedimentation profile of the free RNA. In *Figure 3A*, the free Ψ RNA is a single peak centered on fraction 6. Addition of 3 μM Δ16–99 Gag (red curve) shifts the majority of this RNA to fraction 8, with a significant tail extending nearly to the bottom of the gradient. When 7.5 μM or higher concentrations of Δ16–99 Gag are added, nearly all the RNA is shifted to a broad peak centered around fraction 13. Qualitatively similar results were obtained with MBSM second Gen (*Figure 3B*) and Rev Comp (*Figure 3C*) RNAs.

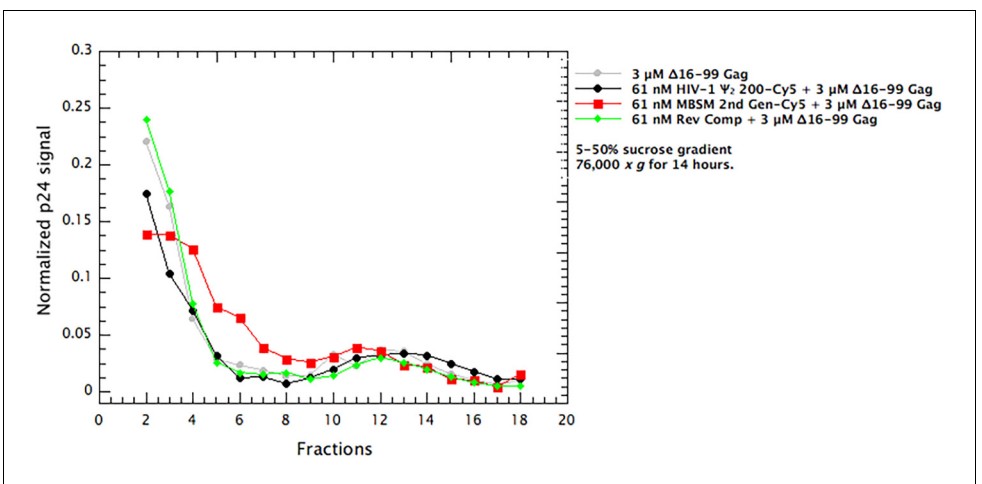

**Figure 5.** Distribution of Δ16–99 Gag in the gradients in *Figure 3A–C*. Aliquots of the gradient fractions were spotted on membranes and treated as in immunoblotting. A parallel dilution series showed that the measurements were within the linear range of the assay. The values are the means of two independent experiments.

DOI: https://doi.org/10.7554/eLife.38438.008

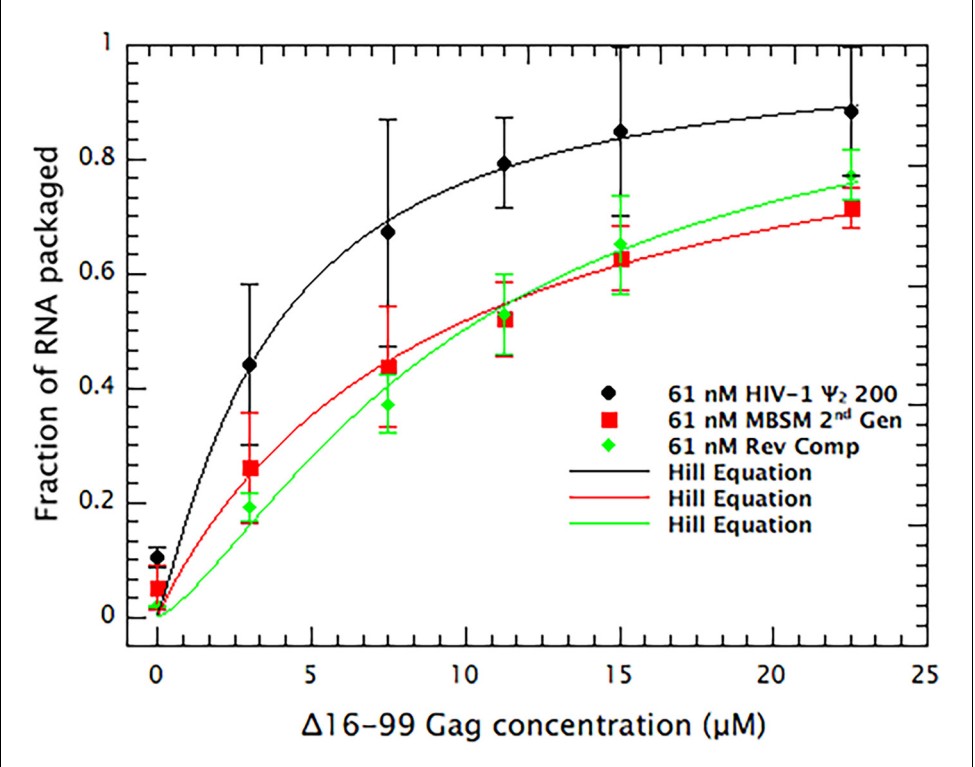

**Figure 6.** Quantitative comparison of the ability of dimeric HIV Ψ 200, MBSM second generation, and Reverse Complement RNAs to support VLP assembly. RNA in fractions 10–18 in **Figure 3(A–C** is summed and plotted *vs.* the concentration of Δ16–99 Gag protein in the assembly reaction. The points are fitted with a cooperative model.
DOI: https://doi.org/10.7554/eLife.38438.009

We also determined the distribution of the Δ16–99 Gag protein in these gradients, by performing immunoblotting on dot blots of aliquots of the gradient fractions (*Figure 5*). We found that in all cases, the vast majority of the protein remained near the top of the gradient (fractions 2–4), and the presence of 61 nM RNA had little or no significant effect upon the distribution of the protein. The fact that the overall protein profile was not significantly affected by the presence of the RNA is not surprising, as the protein was in 50-fold molar excess over the RNA in these gradients.

In order to quantitatively assess the level of VLP assembly in each of the reactions, we summed the amount of RNA between fractions 10 and 18. The results of this analysis are shown in *Figure 6*. These data were fitted, using the non-linear least squares Levenberg-Marquardt method, using the equation

$$Y(x) = \frac{1}{1 + \left(\frac{K}{X}\right)^n}$$

where x is the protein concentration, Y(x) is the fraction of RNA in the bottom half of the tube, and n is a fitting parameter. This equation is analogous to the Hill cooperative model for macromolecular association. Solving for these values yielded the results shown in *Table 2*.

The results reveal a striking difference between Ψ RNA and either MBSM second Gen or Rev Comp RNA: particularly at the lower protein levels, Ψ supports assembly far more efficiently than the other RNAs. For example, at 11.25 μM Δ16–99 Gag, approximately ⁴/₅ of the Ψ RNA has been shifted into the bottom half of the gradient, while only about half of the MBSM second Gen or Rev Comp RNA has undergone a similar shift.

These results are in complete concordance with our hypothesis that binding to a packaging signal nucleates assembly with particularly high efficiency (*Comas-Garcia et al., 2016*; *Nikolaitchik et al., 2013*). Simulations by Perlmutter and Hagan (*Perlmutter and Hagan, 2015*) also demonstrate the

**Table 2.** K and n values and their errors from data shown in *Figure 6*.

| Sample | K (µM) | Error (µM) | N | Error |
|---|---|---|---|---|
| HIV Ψ 200 | 3.74 | ±0.56 | 1.2 | ±0.2 |
| MBSM 2nd Gen | 9.34 | ±0.58 | 1.0 | ±0.1 |
| Rev Comp | 9.94 | ±0.42 | 1.3 | ±0.1 |

DOI: https://doi.org/10.7554/eLife.38438.010

quantitative plausibility of this hypothesis. The fact that when Gag is limiting, there is more assembly on Ψ than on other RNAs (shown here in a defined system in vitro), has also been demonstrated in vivo (*Dilley et al., 2017*); our finding that the same result is obtained in a defined in vitro system shows that this is a direct reflection of the interactions between Gag and the RNAs, and that other cellular components do not drive this phenomenon to any significant degree. The second important finding presented here is that the unpaired guanines within the first few hundred bases of HIV-1 RNA make a major contribution to the specific interactions between Gag and Ψ, as manifested in direct binding assays (*Figure 1*). In fact, the contribution of these clusters of unpaired bases is far more important than that of either SL1 or SL3. Somewhat similar data have been reported by Webb et al. (*Webb et al., 2013*). Furthermore, these unpaired bases are critical for efficient VLP assembly, under conditions in which the protein binds equally well to all the RNAs tested (*Figures 3,6*). Altogether, these results support our hypothesis that Ψ promotes selective packaging of the HIV-1 genomic RNA by virtue of its distinctive efficiency in promoting particle assembly. The data suggest that binding to Ψ reduces the activation energy of the assembly process. We believe that this phenomenon explains the selective packaging of gRNA, in preference to other, cellular RNAs, into virions in infected cells. Experiments to identify a hypothetical nucleating complex are now under way.

# Materials and methods

**Key resources table**

| Reagent type (species) or resource | Designation | Source or reference | Identifiers | Additional information |
|---|---|---|---|---|
| Recombinant DNA reagent | Δp6 Gag expression plasmid | PMID 9971810 | | |
| Recombinant DNA reagent | Δ16–99 Gag expression plasmid | PMID 10619849 | | |
| Other | Ψ150 RNA | GenBank: AF324493.2 | | nt 150–550 |
| Other | Ψ200 RNA | GenBank: AF324493.2 | | nt 200–600 |
| Other | ΔSL1 RNA | GenBank: AF324493.2 | | nt 150–180 joined to nt 280–650 |
| Other | ΔSL3 RNA | GenBank: AF324493.2 | | nt 150–305 joined to nt 405–650 |
| Other | MBSM first generation RNA | GenBank: AF324493.2 | | G224, G226, G240, G241, C243, G270, G272, G273, C274, G275, G289, G290, G292, G310, C312, G318, G320, G328, G239 of Ψ200 replaced with A's |
| Other | MBSM second generation RNA | GenBank: AF324493.2 | | G442, G443, G444, C445, G448, C449, G451, G452, G453, G455, C456, G459 of MBSM 1 st generation replaced with A's |
| Other | GC loop mutant RNA | GenBank: AF324493.2 | | G442, G443, G444, C445, G448, C449, G451, G452, G453, G455, C456, G459 of Ψ200 replaced with A's |
| Other | Reverse Complement RNA | GenBank: AF324493.2 | | RNA is complementary to Ψ150 |

Except where otherwise specified, all procedures were as previously described (*Comas-Garcia et al., 2017*). RNAs were produced by in vitro transcription of linearized plasmids containing

the T7 promoter. All transcripts were 401 nucleotides long unless indicated otherwise and were ultimately derived from the pNL4-3 molecular clone of HIV-1. Numbering begins with the first nucleotide in the R region, equivalent to nt 454 in the DNA sequence. Specifically, HIV-1 Ψ 150 represents nucleotides 150–550; HIV-1 Ψ 200 contains nt 200–600; ΔSL1 contains nt 150–180 and 280–650; ΔSL3 contains nt 150–305 and 405–650; 1st-generation MBSM was derived from HIV-1 Ψ 200 by replacement of G224, G226, G240, C243, G241, G270, G272, G273, C274, G275, G289, G290, G292, G310, C312, G318, G320, G328, and G329 with adenines (*Wilkinson et al., 2008*). The RNA transcribed from this HIV Ψ 200-derived plasmid would still contain a highly GC-rich sequence which would quite possibly be unpaired. To eliminate this potential source of unpaired G residues, we also generated the MBSM second-generation, in which the first-generation MBSM was modified by replacing G442, G443, G444, C445, G448, C449, G451, G452, G453, G455, C456, and G459 with adenines. This latter series of changes was also produced in HIV-1 Ψ 200, yielding the 'HIV-1 GC loop' plasmid. In some experiments, the negative strand complementary to the HIV-1 Ψ 150 RNA ('Reverse Complement') was produced by transcribing a plasmid in which the T7 promoter was at the 3' end, rather than the 5' end, of the HIV-1 ψ 200 insert. The inserts in all plasmids were completely verified by sequencing.

MST measurements were performed in premium coated capillaries on a Monolith NT.115 instrument according to the manufacturer's instructions (Nanotemper Technologies GmbH). Samples were incubated 20 min at 22°C after loading into measuring capillaries. All experiments were done with temperature control set to 22°C. LED power was 90% for Ψ and second generation MBSM RNAs and 50% for Reverse complement RNA. MST power was 20% for all measurements with 5 s fluorescence read before MST laser on, 20 s MST laser switched on and 5 s fluorescence read after MST laser off.

## Acknowledgements

This study was supported by the Intramural Research Program of the NIH, National Cancer Institute, Center for Cancer Research, and in part with funds from the Intramural AIDS Targeted Antiviral Therapy Program. We thank Karin Musier-Forsyth and Roya Zandi for critical discussions and Sergey Tarasov and Marzena Dyba for help with MST measurements.

## Additional information

### Funding

| Funder | Grant reference number | Author |
|---|---|---|
| National Cancer Institute | | Mauricio Comas-Garcia<br>Tomas Kroupa<br>Siddhartha AK Datta<br>Demetria P Harvin<br>Wei-Shau Hu<br>Alan Rein |
| National Institutes of Health | Intramural AIDS Targeted Antiviral Therapy Program | Mauricio Comas-Garcia<br>Alan Rein |

The funders had no role in study design, data collection and interpretation, or the decision to submit the work for publication.

### Author contributions

Mauricio Comas-Garcia, Conceptualization, Resources, Data curation, Formal analysis, Investigation, Methodology, Writing—original draft, Writing—review and editing; Tomas Kroupa, Data curation, Methodology; Siddhartha AK Datta, Conceptualization, Resources, Data curation, Methodology, Writing—review and editing; Demetria P Harvin, Resources, Investigation, Methodology; Wei-Shau Hu, Conceptualization, Investigation, Writing—review and editing; Alan Rein, Conceptualization, Supervision, Funding acquisition, Methodology, Writing—original draft, Writing—review and editing

Author ORCIDs

Mauricio Comas-Garcia (iD) http://orcid.org/0000-0002-7733-5138
Tomas Kroupa (iD) https://orcid.org/0000-0002-5996-9057
Alan Rein (iD) http://orcid.org/0000-0002-8273-546X

Decision letter and Author response
Decision letter https://doi.org/10.7554/eLife.38438.013
Author response https://doi.org/10.7554/eLife.38438.014

## Additional files

**Supplementary files**
• Transparent reporting form
DOI: https://doi.org/10.7554/eLife.38438.011

**Data availability**

All data generated during this study are included in the manuscript and supporting files.

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
