## [Decision Letter]

Thank you for submitting your article "Efficient support of virus-like particle assembly by the HIV-1 packaging signal" for consideration by *eLife*. Your article has been reviewed by three peer reviewers, one of whom is a member of our Board of Reviewing Editors, and the evaluation has been overseen by Gisela Storz as the Senior Editor. The following individual involved in review of your submission has agreed to reveal his identity: Jeremy Luban (Reviewer #3).

The reviewers have discussed the reviews with one another and the Reviewing Editor has drafted this decision to help you prepare a revised submission.

Summary:

This paper, submitted as a Research Advance, presents additional information about the interactions between the HIV-1 genomic RNA and Gag protein constructs. The work extends the earlier results demonstrating both specific and nonspecific RNA binding activity by Gag, and the need for either high salt or high competitor RNA concentrations to reveal the specific binding. (The earlier paper includes much more theoretical analysis of the binding, and modeling of the electrostatics involved.) Based on the earlier work, most readers might have come to the conclusion that in vivo, high competitor RNA levels were present and could have explained the high efficiency of viral RNA packaging. However, the authors here provide additional support for their proposal that the HIV-1 RNA triggers virion assembly more efficiently than other RNAs and that this explains the higher abundance of viral RNA in the final virion preps. Here the binding curves are similar to before (Figure 2). The newer part is the sedimentation of assembled Gag constructs in the presence of RNA, measuring the rapid sedimentation resulting from coassembly (Figure 3). The specific packageable RNA is shown to induce particles at lower concentration than the nonspecific RNAs, and analysis of the Hill equation for binding suggests about a three-fold lower K value. While the electron micrographs of particles are underwhelming, the RNA results (Figure 4 and Table 1) seem to confirm the claim of selective assembly. The punch line should be of interest to the community. The new RNA mutants also provide some new information about what is required for the enhanced assembly.

One reviewer had some specific issues that need to be addressed. Their review is included in its entirety below. Their points are valid and deserve as detailed a response as possible.

*Reviewer #2:*

The *eLife* manuscript by Comas-Garcia et al. is a Research Advance submission based on the previous publication of Comas-Garcia et al., entitled "Dissection of specific binding of HIV-1 Gag to the 'packaging signal' in viral RNA." The stated advances of the current manuscript are support for the hypothesis that encapsidation signal positive (Ψ+) RNAs initiate particle assembly more efficiently than other RNAs, and that unpaired G residues are important elements of the Ψ signal. The degree to which these observations are novel enough to support *eLife* publication, rather than publication in a more specialized journal, is a matter of question. The certainty of the results, especially as regards particle assembly initiation, also is a concern. Specific comments are as follows:

1) Figures 1 and 2A: The results concerning the potential importance of unpaired G residues to RNA binding appear reasonably solid, but have been reported in other experimental systems already (Keane et al., 2015; Rye-McCurdy et al., 2016). I also am troubled by the statement in the legend to Figure 1 that results are the means of only two independent experiments, and that "experiments giving K_d_ values differing by >10% from the consensus values were discarded." It would be nice to know how many such experiments were discarded, and why they were deemed inaccurate.

2) MST experiments: My understanding is that MST experiments were not performed in the initial studies. Because of this, presenting some of the raw data on controls is of interest, as would be showing results with additional RNA substrates, and giving the actual Hill coefficients.

3) Ψ+ RNA effects on assembly: The hypothesis that Ψ+ RNA triggers particle assembly is a logical one, and has received support elsewhere. With respect to the current manuscript, the proof for the hypothesis has not been explained clearly. As I understand it, the crux of the proof here is that the Ψ 2 200 RNA binds Δ16-99 Gag in assembly buffer with the same affinity as the other RNAs (Figure 2B), but partitions more efficiently into higher S value fractions when assembled with higher concentrations of Δ16-99 Gag (Figure 3). The approach and results raise some questions and concerns:

a) Why did the authors have to use Δ16-99 Gag rather than the otherwise wild type Δp6 Gag?

b) Why doesn't Δ16-99 Gag quench the fluorophore in FCS experiments?

c) The Δ16-99 Gag protein binds significantly less well to RNAs at 150 mM NaCl than the Δp6 Gag protein (Figure 2). What happens when the ionic strength is raised, or yeast RNA is added?

d) Figure 3: Please show deviations on the graphs. Please also state how many experiments were excluded because the positions of the peaks were different from the consensus profiles.

e) Figure 3—figure supplement 1: This should be in the paper. I also see lots of spots in A, things that look like tiny rods in B, and a clean background in C. The particles in C also look smaller than in A and B. Please explain.

f) Figure 3—figure supplement 2: This should be in the paper. It also would be useful to see the Gag profiles for the incubations with all the higher concentrations of Gag. It would have been nice to have seen a shift in Gag.

g) Figure 3: What happens when an excess of yeast tRNA is added?

---

## [Author Response]

One reviewer had some specific issues that need to be addressed. Their review is included in its entirety below. Their points are valid and deserve as detailed a response as possible.Reviewer #2:The eLife manuscript by Comas-Garcia et al. is a Research Advance submission based on the previous publication of Comas-Garcia et al., entitled "Dissection of specific binding of HIV-1 Gag to the 'packaging signal' in viral RNA." The stated advances of the current manuscript are support for the hypothesis that encapsidation signal positive (Ψ +) RNAs initiate particle assembly more efficiently than other RNAs, and that unpaired G residues are important elements of the Ψ signal. The degree to which these observations are novel enough to support eLife publication, rather than publication in a more specialized journal, is a matter of question. The certainty of the results, especially as regards particle assembly initiation, also is a concern. Specific comments are as follows:1) Figures 1 and 2A: The results concerning the potential importance of unpaired G residues to RNA binding appear reasonably solid, but have been reported in other experimental systems already (Keane et al., 2015; Rye-McCurdy et al., 2016).

The reviewer notes that previous publications have also highlighted the importance of unpaired G residues to RNA binding, as we mentioned in the original manuscript. However, we would point out that the previous studies involved binding of free NC protein (not Gag) to RNA (Keane et al., 2015) and RNA packaging in vivo (also Keane et al., 2015). We previously showed that both MA and CA domains make important contributions to RNA-binding (Comas-Garcia et al., 2017), so that binding by Gag is not equivalent to binding by NC protein. Effects on packaging in vivo could, of course, be mediated or influenced by any cellular constituent; in contrast, our results show directly that the G’s are important in the salt-resistance of the binding of Gag to Ψ and in the efficiency with which Ψ supports in vitro assembly, in the absence of any other reactants. In addition, Webb et al., 2013(this would have been a more appropriate citation than Rye-McCurdy et al., 2016 and we have now replaced that reference with Webb et al., 2013 in the revised manuscript) did report that the binding of Δp6 Gag to a Ψ construct was less salt-resistant if 12 bases, mainly G’s, were changed to A’s; this “Psi-12M” is similar to our “MBSM” RNA. While the general conclusions about binding in Webb et al. are similar to those in our Figure 1D, we would also note that Webb et al. used a different experimental approach from ours: rather than assaying binding directly at different ionic strengths, they monitored dissociation of preformed protein-RNA complexes as the ionic strength was raised. As we discussed briefly in our 2017 paper, this approach assumes a linear relationship between [Na^+^] and K_D_, and could not have detected the change in the proteins’ binding properties as the salt concentration is raised. We have now pointed out their prior results in the revised manuscript (Introduction, fourth paragraph).

I also am troubled by the statement in the legend to Figure 1 that results are the means of only two independent experiments, and that "experiments giving K_d_ values differing by >10% from the consensus values were discarded." It would be nice to know how many such experiments were discarded, and why they were deemed inaccurate.

The reviewer also asks about experiments that were discarded, as mentioned in the legend to Figure 1. Only two experiments were discarded. In one, an attempt to measure binding of GagΔp6 to 2^nd^ generation MBSM RNA at 0.4M NaCl, no binding was detected, in stark contrast to other experiments. We suspect that the target RNA had been degraded. The other was a competition experiment attempting to measure binding to RevComp RNA in the presence of tRNA. This experiment was discarded because the binding curve failed to reach a plateau.

2) MST experiments: My understanding is that MST experiments were not performed in the initial studies. Because of this, presenting some of the raw data on controls is of interest, as would be showing results with additional RNA substrates, and giving the actual Hill coefficients.

The reviewer asks that more data be included on MST, which was used in the present manuscript to measure binding of protein to RNAs but was not used previously. We have added Figure 2—figure supplement 1, which presents in panels A and B, the raw MST time-trace data for binding of Δ16-99 Gag to ψ RNA in 0.15 and 0.5M NaCl; in panel C, the superimposed cross-sectional profiles of the fluorescence in the MST capillaries, where the symmetrical curves indicate that nothing is sticking to the walls of the capillaries; and in panel D, the profiles of the individual capillaries (containing RNA + different amounts of protein); the near-identity of these profiles shows that the capillaries all contained same amount of fluorescent RNA. The results of the MST measurements are now summarized in Table 1.

3) Ψ + RNA effects on assembly: The hypothesis that Ψ + RNA triggers particle assembly is a logical one, and has received support elsewhere. With respect to the current manuscript, the proof for the hypothesis has not been explained clearly. As I understand it, the crux of the proof here is that the Ψ 2 200 RNA binds Δ16-99 Gag in assembly buffer with the same affinity as the other RNAs (Figure 2B), but partitions more efficiently into higher S value fractions when assembled with higher concentrations of Δ16-99 Gag (Figure 3). The approach and results raise some questions and concerns:a) Why did the authors have to use Δ16-99 Gag rather than the otherwise wild type Δp6 Gag?

We are sorry that we did not explain this in the original submission. We found many years ago (Campbell and Rein, 1999) that the VLPs formed when nucleic acid is added to GagΔp6 are far smaller than authentic HIV-1 virions, and thus cannot have the same architecture as authentic immature particles. In contrast, Δ16-99 Gag forms particles of the correct size in the presence of nucleic acid, and extensive studies from several labs have shown that the overall architecture of these VLPs is an excellent facsimile of that in immature particles (Campbell et al., 2001; Briggs et al., 1993; Gross et al., 2001; Wilk et al., 2001). We have added a brief explanation to the revised manuscript (Results and Discussion, third paragraph).

b) Why doesn't Δ16-99 Gag quench the fluorophore in FCS experiments?

We do not know. As we discussed in the original 2017 manuscript, the quenching of Cy5 by GagΔp6 represents stabilization of a non-fluorescent isomer of Cy5. We have also observed that free NC protein does not quench Cy5-labeled RNAs. These observations suggest that the quenching involves the MA domain in GagΔp6, but we have no further information on this. We have noted this in the fourth paragraph of the Results and Discussion.

c) The Δ16-99 Gag protein binds significantly less well to RNAs at 150 mM NaCl than the Δp6 Gag protein (Figure 2). What happens when the ionic strength is raised, or yeast RNA is added?

We are now investigating the role of the MA domain in specific and non-specific RNA-binding. We have found that the binding of Δ16-99 Gag to Ψ RNA, as well as to the control RNAs, is strongly depressed at high ionic strengths. We have not yet tested the effect of competitor tRNA. As mentioned above, we have added a supplementary figure and a Table with the MST data for binding of Δ16-99 Gag to the RNAs.

d) Figure 3: Please show deviations on the graphs. Please also state how many experiments were excluded because the positions of the peaks were different from the consensus profiles.

We have added the deviations to Figure 3 (now Figure 4) as requested by the reviewer. Two experiments were discarded because the labeled RNA barely moved from the top of the gradient. We assume that the RNA in these samples was degraded.

e) Figure 3—figure supplement 1: This should be in the paper. I also see lots of spots in A, things that look like tiny rods in B, and a clean background in C. The particles in C also look smaller than in A and B. Please explain.

The reviewer asks that the electron micrographs, which were Figure 3—figure supplement 1 in the original manuscript, be moved to the body of the paper. We have now made them Figure 3. The reviewer also asks for a little more information about what is seen in the electron micrographs. Our intention was just to demonstrate that the protein successfully assembles into VLPs under the conditions of these experiments. We have also added an inset to each of the 3 panels, presenting a VLP at higher magnification for greater clarity. Finally, the reviewer asks about the miscellaneous structures in these images which are not spherical VLPs. It is obvious that a wide variety of structures are formed and we have noted that in the revised manuscript (Results and Discussion, sixth paragraph). Finally, the reviewer mentions a “clean background” in C. We have not investigated this systematically as yet, but the reviewer may well be correct that fewer structures other than intact VLPs are formed on Reverse Complement RNA. We have noted this possibility in the aforementioned paragraph.

f) Figure 3—figure supplement 2: This should be in the paper. It also would be useful to see the Gag profiles for the incubations with all the higher concentrations of Gag. It would have been nice to have seen a shift in Gag.

The reviewer asks that Figure 3—figure supplement 2, the profile of Gag in the sucrose gradients, be included in the paper. We have now made it Figure 5. The reviewer also asks about the profiles obtained with higher Gag concentrations, where Gag is in even greater excess over the RNA. In fact the amounts of Gag here were so high that it is not trivial to obtain quantitative data on Gag in these gradients. The vast majority of Gag remains at the top of the gradient; this does not seem like useful information to us and we would prefer to omit these profiles.

g) Figure 3: What happens when an excess of yeast tRNA is added?

The reviewer asks about adding competitor RNA, such as yeast tRNA, in the assembly experiments. We agree that this is an interesting suggestion but have not tried it as yet.